# Three Decades after: Landscape Dynamics in Different Colonisation Models Implemented in the Brazilian Legal Amazon

Valdir Moura [1,2,*], Ranieli dos Anjos de Souza [1], Erivelto Mercante [2], Jonathan Richetti [3] and Jerry Adriani Johann [2]

1. Space Research Group (GREES), Federal Institute of Rondônia (IFRO), Colorado do Oeste 76930-000, Brazil; ranieli.anjos@ifro.edu.br
2. Graduate Program in Agricultural Engineering (PGEAGRI), State University of Western Paraná (UNIOESTE), Cascavel 85819-110, Brazil; eriveltomercante@yahoo.com.br (E.M.); jerry.johann@hotmail.com (J.A.J.)
3. CSIRO Agriculture & Food, Private Bag 5, Wembley, WA 6014, Australia; jonathan.richetti@csiro.au
* Correspondence: valdir.moura@ifro.edu.br; Tel.: +55-69-999208560

**Abstract:** Several colonisation projects were implemented in the Brazilian Legal Amazon in the 1970s and 1980s. Among these colonisation projects, the most prominent were those with the "fishbone" and "topographic" models. Within this scope, the settlements known as Anari and Machadinho stand out because they are contiguous areas with different models and structures of occupation and colonisation. The main objective of this work was to evaluate the dynamics of Land-Use and Land-Cover (LULC) in two different colonisation models, implanted in the State of Rondônia in the 1980s. The fishbone and topographic or Disorganised Multidirectional models were implemented in the Anari and Machadinho settlements, respectively. A 36-year time series of Landsat images (1984–2020) was used to evaluate the rates and trends in the LULC process in the different colonisation models. In the analysed models, a rapid loss of primary and secondary forests (anthropized areas) was observed, mainly due to the dynamics of its use, established by the Agriculture/Pasture relation with a heavy dependence on road construction. Understanding these two forms of occupation can help the future programs and guidelines of the Brazilian Legal Amazon and any tropical rainforest across the globe.

**Keywords:** deforestation; environmental; rate fragments; remote sensing; secondary succession; anthropized areas

## 1. Introduction

The process of occupation of the Brazilian Legal Amazon (BLA) has several levels. The first level refers to access to title deeds, the trade of these titles and, therefore, the involvement of social groups who hold power in the state [1,2]. This has resulted in the emergence of corruption mechanisms involving different access routes to land. Therefore, agricultural and agribusiness projects financed by the Superintendence for the Development of the Amazon (SUDAM) largely consist of losses against public funds [3,4]. Subsequently, agrarian reform (the distribution of rural property by the Brazilian government) has implemented private and public settlement projects. Following this, an increase in logging, agriculture, and agribusiness was observed [5,6]. All these processes resulted in violence against indigenous people, settlers, miners, pedestrians, and especially against nature [7,8], further increasing deforestation [9].

Deforestation can be defined as the process of transformation or change in an area of primary or secondary forest that leads to the replacement of the original type of land cover by another, either immediately or progressively [10–12]. A common approach to studying deforestation is to consider it as a binary process in which possible forms of land cover are non-forest and forest [13,14]. However, the dynamic of Land-Use and Land-Cover

(LULC) includes the regeneration process as well as the loss of coverage; the net result is the subtraction and addition of derivatives of both tendencies [15,16]. A growing number of studies of LULC dynamics have considered this balance between loss and regeneration, particularly in areas of high environmental and socioeconomic heterogeneity [17–19].

High rates of deforestation are of major worldwide concern, especially regarding the future of the Amazon rainforest. Concerns include projections of drastic changes in the original landscape over the next 20 years if current rates of deforestation, development, and infrastructure projects in the region are maintained [20–22]. Thus, future changes in land use in the BLA must present new paradigms that seek sustainability [23,24].

Although changes in land use are complex and often multidirectional interactions of biophysical and socioeconomic factors [22,25,26], they are, fundamentally, local processes tied to a hierarchical decision-making structure [4,18,27]. Land cover patterns are observed by the outcome of each decision-making process regarding land use return [6,28] and after stabilisation within a regional or national context. Thus, land cover, along with the analysis of patterns and measures related to social sciences, can be used to indicate changes in land use patterns. Additionally, the BLA rainforest has a global impact.

This study aims to evaluate the dynamics of LULC in two different colonisation models implemented in the BLA state of Rondônia in the 1980s. The objective is to evaluate the dynamics of deforestation in the Machadinho and Anari settlement projects located in the northeast of the state of Rondônia, emphasising anthropic actions. It is hypothesised that a nature-friendly colonisation model can be more sustainable over time when compared to a colonisation model that disregards the surrounding environment. Thus, a comparison between a topographic (in Machadinho) and a fishbone (in Anari) colonisation models was performed. We used 36 years of Landsat data associated with the field analysis (interviews), establishing the rates, patterns of land tenure dynamics and the remaining fragments within the perimeter of the two colonisation projects, as well as providing a methodology that effectively contributes to the evaluation of the multitemporal dynamics of land use and coverage through regional-scale analysis for the different colonisation projects implemented in the state of Rondônia.

## 2. Materials and Methods

### 2.1. Study Area

The area used in the study is composed of the settlement projects named Anari and Machadinho. These projects originated in the Vale do Anari and Machadinho do Oeste municipalities, respectively, both located in the northeast of the state of Rondônia (Figure 1). With a total area of 702.48 and 2129.98 km$^2$, respectively, these settlement projects correspond to 22.41% and 25.35% of the area defined by law for the two municipalities in question. The locations are neighbouring municipalities and share the same environmental conditions. They are also both in the tropical climate, are located about 400 km from the capital Porto Velho, and are accessible by road, while river transport can be particularly useful during the rainy season. The dense rainforest is the predominant natural vegetation [29]. Conservation reserves are planned throughout the area and have varying degrees of protection depending on the level of human intervention. The settlement projects under study present different models of occupation, with the fishbone model being implemented in the Anari and the topographic or disorganised multidirectional model in the Machadinho.

The fishbone model was materialized, having as a reference the roads built horizontally up to the limits of the settlement, placed between them for a distance of 4 km. From these roads, the limit of the lots were inserted. The lots had characteristics of 250 m × 2000 m. In this model, the drainage mesh was not observed during the implementation of the lots. In the topographic model, the natural limits were used as a reference in the implementation of the lots, whose characteristics (size and shape) were irregular.

In 2016, 280 questionnaires and interviews with local settlers were conducted to understand the motivations and reasoning behind the LULC changes. In 2021, all points

where the questionnaires had been applied in 2016 were verified in loco within the two settlement projects.

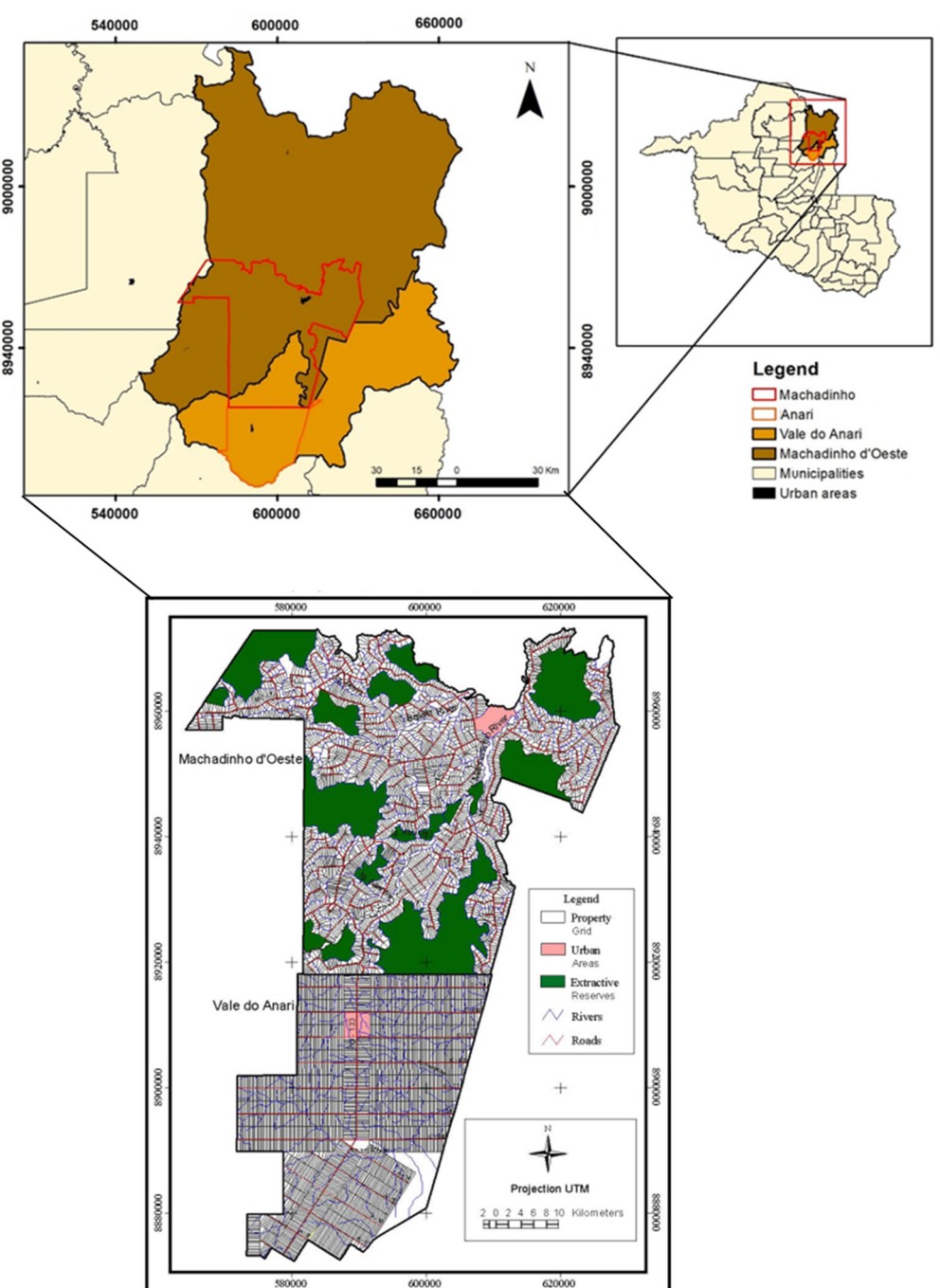

**Figure 1.** Political division of the study area, with emphasis on the municipalities of Machadinho D'Oeste and Vale do Anari and the settlements Machadinho and Anari in the Brazilian Legal Amazon.

### 2.2. Satellite Data and LULC Mapping

Due to a long time series, two different orbital data sets (Landsat TM and OLI) orbits 231/66 and 231/67 were used. The TM data (B3, B4 and B5) were used for from 1984 to 2011. The OLI data (B4, B5 and B6) were used from 2013 to 2020. The orbital data used are spatially and spectrally equivalent. Images were acquired in June, July or August of each year as follows: 1984 (4 August 1984), 1989 (17 July 1989), 1994 (17 July 1989), 1999 (29 July

1999), 2004 (26 July 2004), 2010 (27 July 2010), 2016 (11 July 2016) and 2020 (22 July 2020). The use of the images acquired during this period ensured cloudless images.

Due to the differences in sensor radiometric specifications, two different random forest models were used. One for Landsat TM with the number of variables randomly sampled as candidates at each split (mtry) of 3 and the number of trees (ntrees) of 200, and a second for Landsat OLI with mtry of 3 and ntrees = 500. To obtain these parameters, a grid search with a range ntrees (100 to 1000) was performed. For each period evaluated, an error matrix was obtained in order to assess the accuracy of the classification.

The Random Forest algorithm [30] was used to map the LULC for each studied location. Eight classes (i) Initial Successional Stage (SSI), (ii) Advanced Successional Stage (SSA), (iii) Agriculture, (iv) Pasture, (v) Bared Soil, (vi) Forest, (vii) Water, and (viii) Infrastructure were classified. A total of 2480 samples were used to train and validate the model in a random 70–30 slipt. That is, 70% (1736 samples) were randomly selected and used for training the model, and the remaining 30% (744 samples) were used to validate the LULC maps. The common practice confusion matrix and global accuracy, kappa index, and producer accuracy were calculated with the validation set.

The 2016 image served as the basis for obtaining the polygons of different land cover classes. These polygons were labelled according to their cover class and subsequently adjusted for each year of analysis. The polygon adjustments were important because they reflected the changes in land cover that took place from 1984 to 2020. This procedure was adopted due to the difficulty in defining the uses at older dates. It made it possible to update the changes that occurred in different periods and avoid false changes that occurred due to the similarity in spectral responses in different scenes [31,32]. The LULC maps were evaluated by periods, 1984–1989, 1989–1994, 1994–1999, 1999–2004, 2004–2010, 2010–2016 and 2016–2020, resulting in eight land cover maps. The eight land cover maps formed the time-series for each colonisation model and were the basis for the analysis of the LULC dynamics (Figure 2).

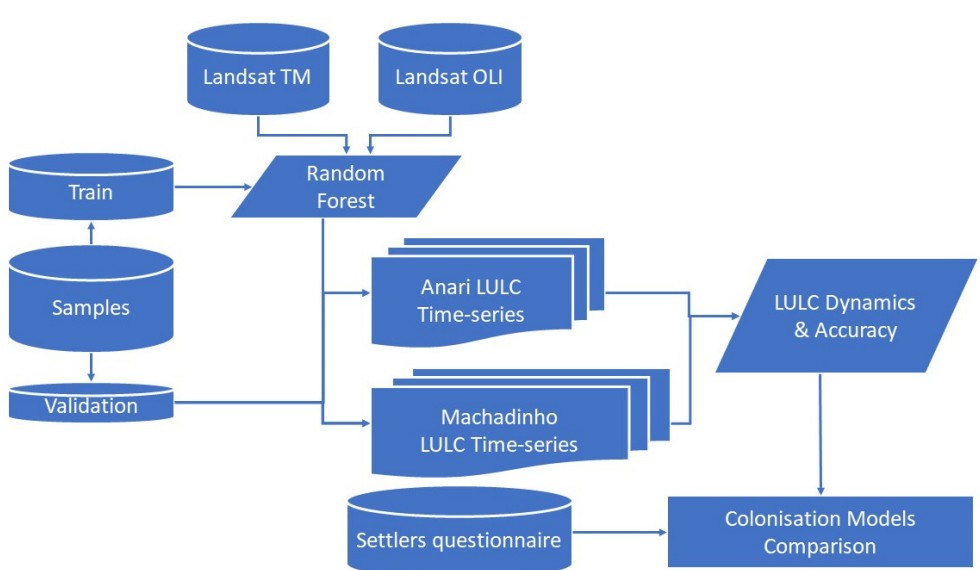

**Figure 2.** Methodology used in this research.

Classes of secondary vegetation (succession) arise after the removal of primary vegetation. This class was used to characterize the deforested places occupied with another class of interest (e.g., pasture) and later, due to the chemical, physical and topographic characteristics of the soil, they were abandoned. In this way, the "succession" class was subdivided into initial (SSI) and advanced (SSA). Where the SSI class was characterized by areas abandoned for a period of less than 7 years. The SSA class, on the other hand, was characterized by the finding of more than 7 years of abandonment. It is noteworthy

that currently, according to the environmental laws in force in the state of Rondonia, after 5 years of abandonment, the area is already classified as an "area in an advanced stage of succession".

### 2.3. LULC Dynamics Analysis—Transition Probabilities and Anthropised Areas Rates

Probabilistic transition matrices were calculated for each of the seven study periods following the methodology by [33]. Briefly, each matrix represents the probability of persistence of each land cover category, varying from the first to the last year of the period, or may represent the transition probabilities occurring between land use and land cover categories over the same period. The matrices values were standardised as recommended by [34], so that the annual variations due to the changes that occur in the use and occupation of the land in the settlement projects were corrected and comparable.

After defining the different coverage classes for the seven periods analysed, the probability matrices were constructed for the periods 1984–1989, 1989–1994, 1994–1999, 1999–2004, 2004–2010, 2010–2016 and 2016–2020 (Appendix A). These matrices represent the probability of remaining in the same class, or the probability of transitioning to another category during the period analysed, according to the LULC dynamics. The values of the matrices were standardised so that the change values were annualised [34]. The matrix standardisation procedure for evaluating land cover change was proposed by [35]. As the time intervals ranged from 4 to 6 years, it was necessary to annualise them. In this procedure, the diagonalisation method proposed by [36] was used, where each probability matrix was separated by calculating the eigenvectors and eigenvalues.

The method adopted in this study assumes that the probability that a sample cell (pixel set) belongs to a certain class $m$ during the initial year of the interval of the analysed period and has presented a change to a class $n$ at the end of the analysed period is defined by Equation (1).

$$r_{mn} = \frac{a_{mn}}{a_m} \tag{1}$$

where $a_{mn}$ is the area covered by class $m$ during the initial year and covered by class $n$ during the final year and $a_m$ the area covered by class $m$ during the initial year

The temporality of the study, as a function of the intervals of years existing between the beginning and the end of the period ($t$ years), determines the probability transition matrix ($R^{(t)}$) by Equation (2).

$$R^{(t)} = [r_{mn}] \tag{2}$$

The assembly of the annual probability matrix ($P = [p_{mn}]$), was based on the maps of land use and cover, where $pmn$ is the probability of change from class $m$ to class $n$ during the year (e.g., bared soil for agriculture...agriculture for bared soil), in this scenario, the probability of transition. In this case, we assume that the transition probability has two characteristics: they are stochastic processes, and they are homogeneous in time. Thus, we have that the probability of transition from one class to another is independent, and is expressed by Equation (3).

$$P * P.....P - P^t - R^{(t)} \tag{3}$$

Sequentially, he calculated the annual probabilities ($p_{mn}$) using diagonalisation, through Equations (4)–(6), as described below:

$$P = B * D * B^{-1} \tag{4}$$

where $D$ is a diagonal matrix. Matrix $D$ has the eigenvalues of $P$ in the diagonal. The columns in $B$ consist of the corresponding eigenvectors. It can be shown that the $D^t$ matrix has the $P^t$ values in the major diagonal, then:

$$P^t = B * D^t * B^{-1}, \quad t = 1, 2, \ldots \tag{5}$$

After determining the $R^{(t)}$, we obtain the annual probability matrix $P$, using Equation (6):

$$P = B * \begin{bmatrix} \sqrt[t]{\lambda_1} & 0 & 0 \\ 0 & \sqrt[t]{\lambda_2} & 0 \\ 0 & 0 & \sqrt[t]{\lambda_n} \end{bmatrix} * B^{-1} \qquad (6)$$

Assuming that the LULC conditions were stationary, we used Markov chain models on the annualised matrices to simulate the coverage proportion for the study periods. Markov chains are stochastic processes and can be parameterised by empirically estimating the transition probabilities between discrete states within the observed system [37]. The annualised matrices for each period (1984–1989, 1989–1994, 1994–1999, 1999–2004, 2004–2010, 2010–2016 and 2016–2020) were analysed using a log-linear statistical test to discern whether they were statistically different. The statistical analysis applied in this step is described in detail in Appendix A.

Lastly, the annual deforestation rates (*DR*) for the studied periods (1984–1989; 1994–1989; 1999–1994; 2004–1999; 2004–2010; 2010–2016; 2016–2020) were assessed based on forest cover data. The *DR* was defined as the opposite of the annual rate of change of forest cover (Equation (7)) [38].

$$DR = \left( \frac{1}{t_2 - t_1} \right) \times \ln \left( \frac{A_2}{A_1} \right) * (-1) \qquad (7)$$

where, *DR* is the rate of deforestation (% lost area/year) between the two periods ($t_1$ and $t_2$); $A_1$ and $A_2$ are forest areas between the two periods.

## 3. Results

### 3.1. Land-Use and Land-Cover Changes

A minimal overall accuracy of 83.8% was observed on LULC mapping (Table 1), sufficient accuracy for assessing the temporal LULC dynamics and trends.

**Table 1.** Evaluation of the accuracy of the LULC classification in the Machadinho and Anari settlements.

| Classes LULC | 1984 | | 1989 | | 1994 | | 1999 | | 2004 | | 2010 | | 2016 | | 2020 | |
|---|---|---|---|---|---|---|---|---|---|---|---|---|---|---|---|---|
| | PU | PP | PU | PP | PU | PP | PU | PP | PU | PP | PU | PP | PU | PP | PU | PP |
| Forest | 92.4 | 90.8 | 89.3 | 94.1 | 87.9 | 92.5 | 88.7 | 94.6 | 91.7 | 95.8 | 90.6 | 93.8 | 96.7 | 97.1 | 98.7 | 98.9 |
| SSA | 65.9 | 79.6 | 60.9 | 68.7 | 59.7 | 64.5 | 59.1 | 69.2 | 60.4 | 67.3 | 51.8 | 59.4 | 72.4 | 78.9 | 83.4 | 85.7 |
| SSI | 66.5 | 81.2 | 61.9 | 67.9 | 54.8 | 61.9 | 62.7 | 69.1 | 53.7 | 63.9 | 52.3 | 61.2 | 79.3 | 81.7 | 87.6 | 88.2 |
| Pasture | 89.6 | 78.7 | 90.7 | 86.8 | 92.1 | 87.3 | 91.2 | 89.6 | 90.4 | 85.9 | 90.1 | 83.9 | 95.9 | 90.7 | 93.2 | 95.8 |
| Agriculture | 84.9 | 81.6 | 89.6 | 81.9 | 90.8 | 89.3 | 92.7 | 89.8 | 92.5 | 90.6 | 94.3 | 91.2 | 95.2 | 97.9 | 99.4 | 99.6 |
| Bare Soil | 85.9 | 94.8 | 91.2 | 93.7 | 92.4 | 94.1 | 93.9 | 94.8 | 91.7 | 93.5 | 92.2 | 94.6 | 96.8 | 97.5 | 99.1 | 99.4 |
| Infrastructure | 82.8 | 100 | 87.9 | 100 | 89.7 | 100 | 88.9 | 100 | 90.6 | 100 | 92.3 | 100 | 98.5 | 100 | 98.7 | 100 |
| Water | 91.3 | 100 | 92.3 | 100 | 94.5 | 100 | 91.8 | 92.3 | 92.5 | 100 | 94.2 | 98.1 | 98.2 | 98.7 | 99.5 | 99.6 |
| Accuracy | 85.3 | | 84.8 | | 84.5 | | 85.5 | | 85.0 | | 83.8 | | 91.8 | | 93.4 | |
| Kappa | 79.5 | | 81.1 | | 81.0 | | 81.7 | | 80.8 | | 80.7 | | 90.6 | | 91.8 | |

Where PU: User Accuracy; PP: Precision product.

The Machadinho and Anari settlements revealed marked differences in landscape change (Tables 2 and 3). During the initial phase of implementation of both settlement projects (base year 1984), there was a similar percentage of forest (95%) and anthropized areas (5%). Ten years later (1994), forest cover fell to 65.49% in Machadinho, in contrast to 68.47% in Anari. Thirty-six years after the start of the study (2020), the rates showed alarming numbers. Only 19% of the forest cover remained intact in Anari, while in Machadinho, 42.97% of the forest cover remained (Figure 3A,B).

The evolution of anthropized areas in the fishbone model (Figure 4A) shows that 2004 had a proxy-linear growth behaviour and that later this year, there was smoothing in the curve and consequent reduction in anthropized areas rates. However, in the topographic or disorganised multidirectional model, the anthropized areas are smaller and less aggressive with a smoother growth (Figure 4B). There has been a subsistence occupation until the

year 1999. As of this date, there was a change in regional behaviour and an acceleration in anthropized areas rates, presenting a proxy-linear and accelerated growth of anthropized areas. This advance in anthropized areas was due to real estate speculation, land valuation, and the advancement of agriculture under the areas of the BLA.

**Table 2.** LULC in settlement of Machadinho for the years 1984, 1989, 1994, 1999, 2004, 2010, 2016 and 2020.

| Class LULC | 1984 | | 1989 | | 1994 | | 1999 | | 2004 | | 2010 | | 2016 | | 2020 | |
|---|---|---|---|---|---|---|---|---|---|---|---|---|---|---|---|---|
| | Area (km²) | % | Area (km²) | % | Area (km²) | % | Area (km²) | % | Area (km²) | % | Area (km²) | % | Area (km²) | % | Area (km²) | % |
| For | 2009.95 | 94.36 | 1704.93 | 80.04 | 1395.01 | 65.49 | 1290.01 | 60.56 | 1111.65 | 52.19 | 1007.29 | 47.29 | 946.63 | 44.44 | 915.19 | 42.97 |
| SSA | 52.83 | 2.48 | 69.99 | 3.29 | 139.06 | 6.53 | 141.03 | 6.62 | 194.52 | 9.13 | 117.56 | 5.52 | 132.93 | 6.24 | 118.50 | 5.56 |
| SSI | 0.89 | 0.04 | 2.37 | 0.11 | 89.77 | 4.21 | 92.31 | 4.33 | 51.62 | 2.42 | 135.82 | 6.38 | 153.51 | 7.21 | 148.95 | 6.99 |
| Past | 6.05 | 0.28 | 182.90 | 8.59 | 325.48 | 15.28 | 345.58 | 16.22 | 515.44 | 24.20 | 641.73 | 30.13 | 628.70 | 29.52 | 627.63 | 29.47 |
| Agr | 0.00 | 0.00 | 0.00 | 0.00 | 9.65 | 0.45 | 9.75 | 0.46 | 29.08 | 1.37 | 65.66 | 3.08 | 80.37 | 3.77 | 95.75 | 4.50 |
| BS | 14.55 | 0.68 | 119.77 | 5.62 | 118.67 | 5.57 | 198.3 | 9.31 | 176.06 | 8.27 | 109.70 | 5.15 | 134.7 | 6.32 | 169.97 | 7.98 |
| IR | 0.30 | 0.01 | 2.13 | 0.10 | 3.36 | 0.16 | 4.65 | 0.22 | 6.72 | 0.32 | 8.07 | 0.38 | 8.09 | 0.38 | 9.26 | 0.43 |
| Water | 45.41 | 2.13 | 47.89 | 2.25 | 48.98 | 2.30 | 48.35 | 2.27 | 44.89 | 2.11 | 44.15 | 2.07 | 45.05 | 2.12 | 44.73 | 2.10 |
| Total | 2129.98 | 100 | 2129.98 | 100 | 2129.98 | 100 | 2129.98 | 100 | 2129.98 | 100 | 2129.98 | 100 | 2129.98 | 100 | 2129.98 | 100 |

Where for: forest; SSA: advanced secondary succession; SSI: initial secondary succession; Past: Pasture; Agr: Agriculture; BS: Bare Soil; IR: Infrastructure.

**Table 3.** LULC in the Anari settlement for the years 1984, 1989, 1994, 1999, 2004, 2010, 2016 and 2020.

| Class LULC | 1984 | | 1989 | | 1994 | | 1999 | | 2004 | | 2010 | | 2016 | | 2020 | |
|---|---|---|---|---|---|---|---|---|---|---|---|---|---|---|---|---|
| | Area (km²) | % | Area (km²) | % | Area (km²) | % | Area (km²) | % | Area (km²) | % | Area (km²) | % | Area (km²) | % | Area (km²) | % |
| For | 673.12 | 95.82 | 582.4 | 82.91 | 480.98 | 68.47 | 405.93 | 57.79 | 356.22 | 50.71 | 175.63 | 25.00 | 152.19 | 21.66 | 133.49 | 19.00 |
| SSA | 15.34 | 2.18 | 7.89 | 1.12 | 29.17 | 4.15 | 28.06 | 3.99 | 55.02 | 7.83 | 77.00 | 10.96 | 67.05 | 9.54 | 52.23 | 7.44 |
| SSI | 0.53 | 0.08 | 0.9 | 0.13 | 15.34 | 2.18 | 12.33 | 1.76 | 17.27 | 2.46 | 77.47 | 11.03 | 45.14 | 6.43 | 35,14 | 5.00 |
| Past | 3.77 | 0.54 | 62.21 | 8.86 | 120.79 | 17.19 | 163.43 | 23.26 | 220.08 | 31.33 | 274.53 | 39.08 | 382.51 | 54.45 | 390.27 | 55.56 |
| Agr | 0.00 | 0.00 | 0.00 | 0.00 | 0.12 | 0.02 | 0.43 | 0.06 | 0.65 | 0.09 | 0.7 | 0.10 | 1.92 | 0.27 | 15.67 | 2.23 |
| BS | 7.48 | 1.06 | 46.12 | 6.57 | 52.33 | 7.45 | 87.76 | 12.49 | 48.95 | 6.97 | 91.56 | 13.03 | 47.88 | 6.82 | 69.94 | 9.96 |
| IR | 0.00 | 0.00 | 0.41 | 0.06 | 0.77 | 0.11 | 1.64 | 0.23 | 1.92 | 0.27 | 3.03 | 0.43 | 3.04 | 0.43 | 3.11 | 0.44 |
| Water | 2.24 | 0.32 | 2.55 | 0.36 | 2.98 | 0.42 | 2.9 | 0.41 | 2.37 | 0.34 | 2.56 | 0.36 | 2.75 | 0.39 | 2.63 | 0.37 |
| Total | 702.48 | 100 | 702.48 | 100 | 702.48 | 100 | 702.48 | 100 | 702.48 | 100 | 702.48 | 100 | 702.48 | 100 | 702.48 | 100 |

The average forest area (Figure 5A) shows that both settlements considerably fragmented the forest; however, the topographic model at Machadinho kept more extensive areas of the forest than Anari (Figure 5B). Regarding SSI class, there was an increase in the agricultural class in the Anari settlement in the year 2010. In this case, from 2010 to 2020, the SSI and SSA class reduced (Figure 5C) in the settlements of Anari and Machadinho due to the increase in rural real estate prices. Much of the vegetation recovery area is effectively used for livestock, resulting in a trend to extensification of pasture areas in Anari. The settlements present a tendency in the conversion to different pasture uses, considering the great difficulty obtained to mechanise the area to be cultivated, mainly due to the slope of the land.

In 1984, settlements presented similar rates in relation to agriculture and bare soil classes (Figure 5E), and there were larger variations observed in the years 1989 and 2004 when the Anari settlement showed occupancy peaks that diverged from the town of Machadinho. The Machadinho settlement in 2010 surpassed Anari in the area occupied by agriculture and exposed/bare soil, a tendency also seen in 2016. In the period from 2016 to 2020, there was accelerated growth in the areas of exposed soil and agriculture, due to soybean cropping advancing significantly under areas that were occupied with pastures and in the process of forest succession (Figure 5E). This change was observed in the two occupation models analysed. According to landowners in both settlements, the conditions of agricultural land are better in Machadinho. This was confirmed after an analysis of the official rates, suggesting better management of the land cultivated by the owners.

The values of the pasture (Figure 5D) in Anari showed a similar increase in Machadinho from 1984 to 1994; however, Anari continued to increase the pasture area almost steadily, while Machadinho decreased the pasture area from 1994 to 2004 and increased once more from 2004 onwards. Comparing the two settlements, Anari had a pasture index 26% greater than that of Machadinho.

Deforestation rates changed over the years, with the highest deforestation rates between 1988 and 2004 (Figure 5F). There was a small recovery of forest area and, thus, the negative numbers.

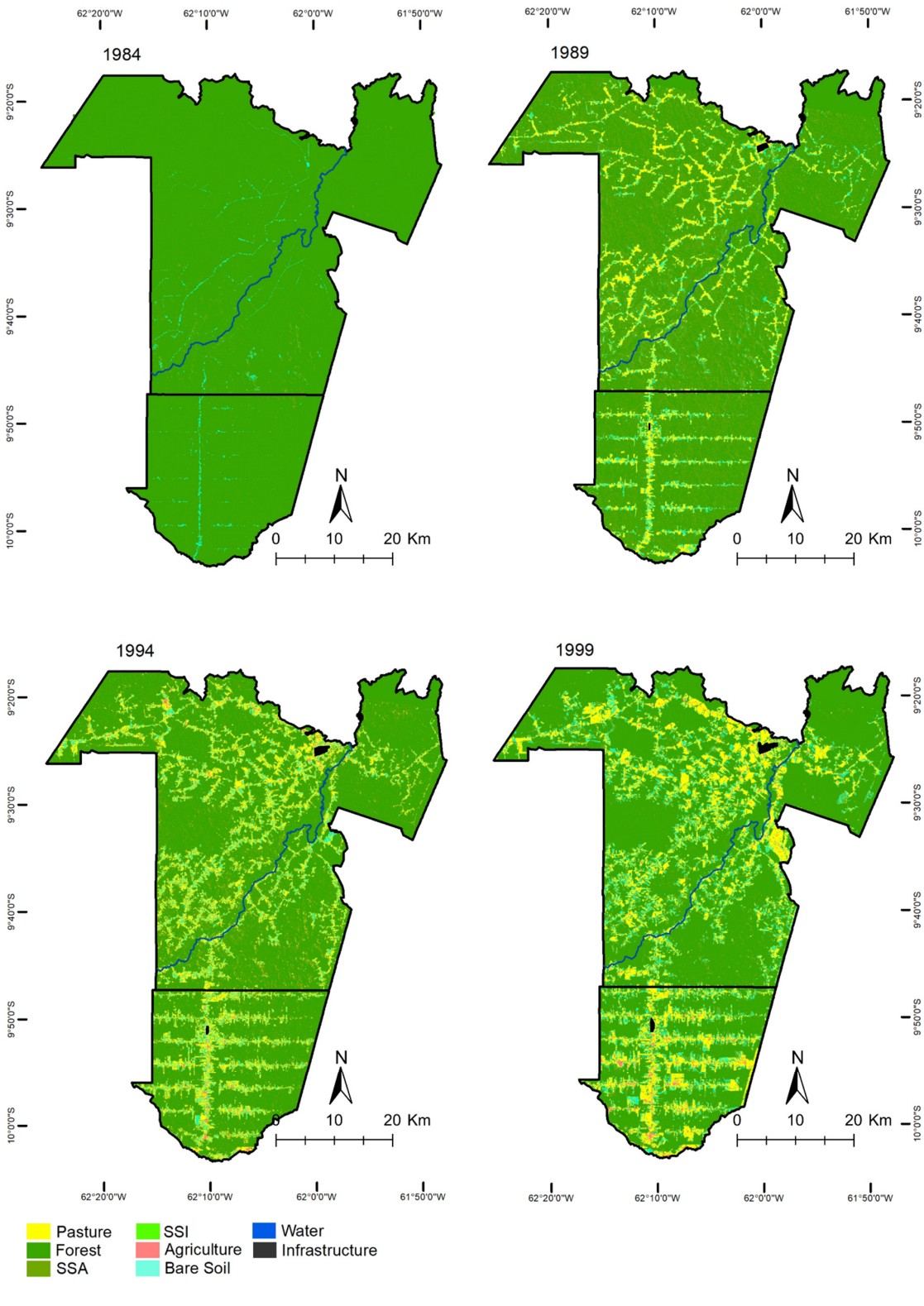

(**A**)

**Figure 3.** *Cont.*

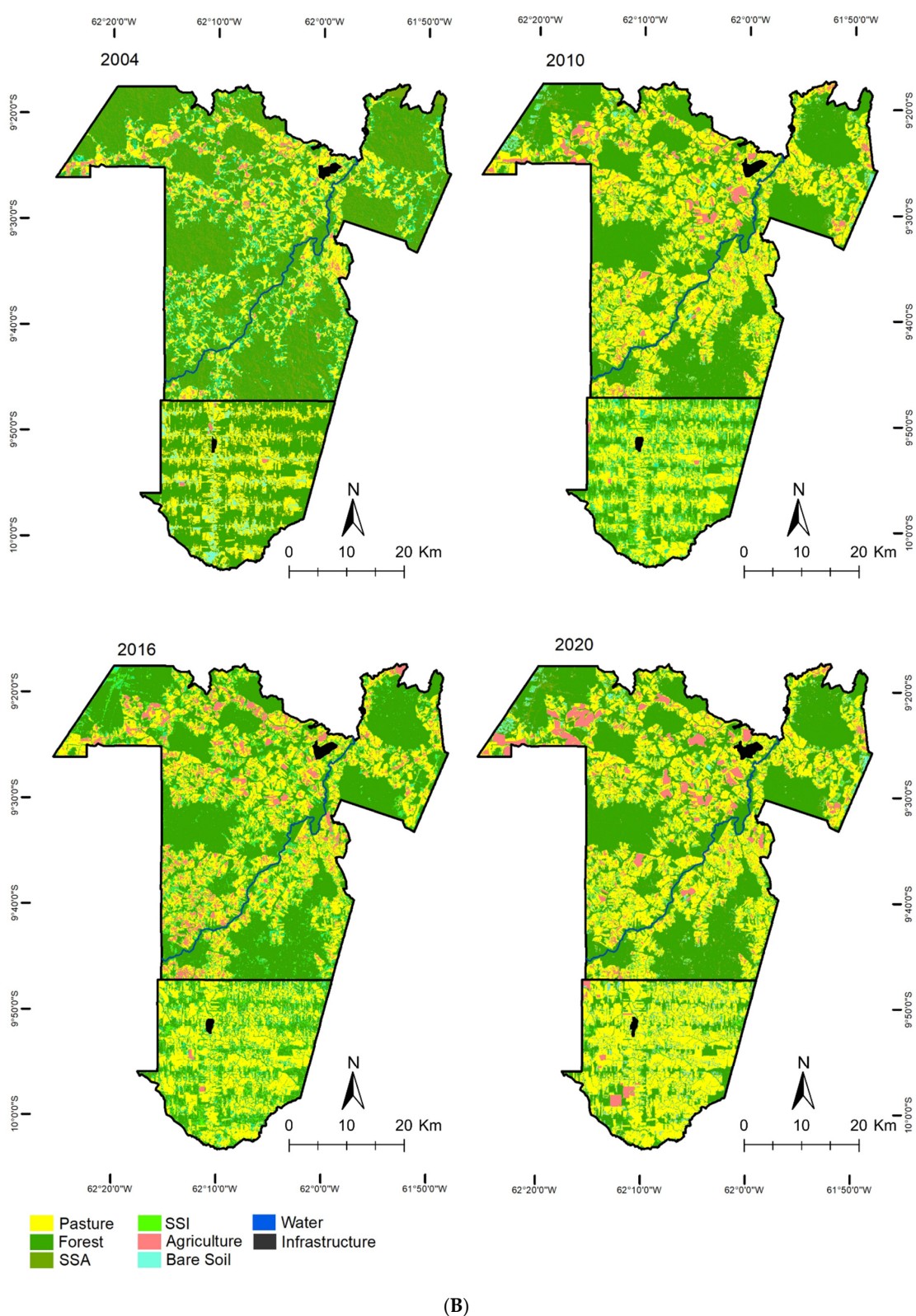

(**B**)

**Figure 3.** (**A**). LULC in the settlements of Machadinho and Anari with open tropical forest (forest), advanced secondary succession (SSA), initial secondary succession (SSI), production (agriculture, pasture, bare soil), water and infrastructure from 1984 to 1999. (**B**). LULC in the settlements of Machadinho and Anari with open tropical forest (forest), advanced secondary succession (SSA), initial secondary succession (SSI), production (agriculture, pasture, bare soil), water and infrastructure from 2004 to 2020.

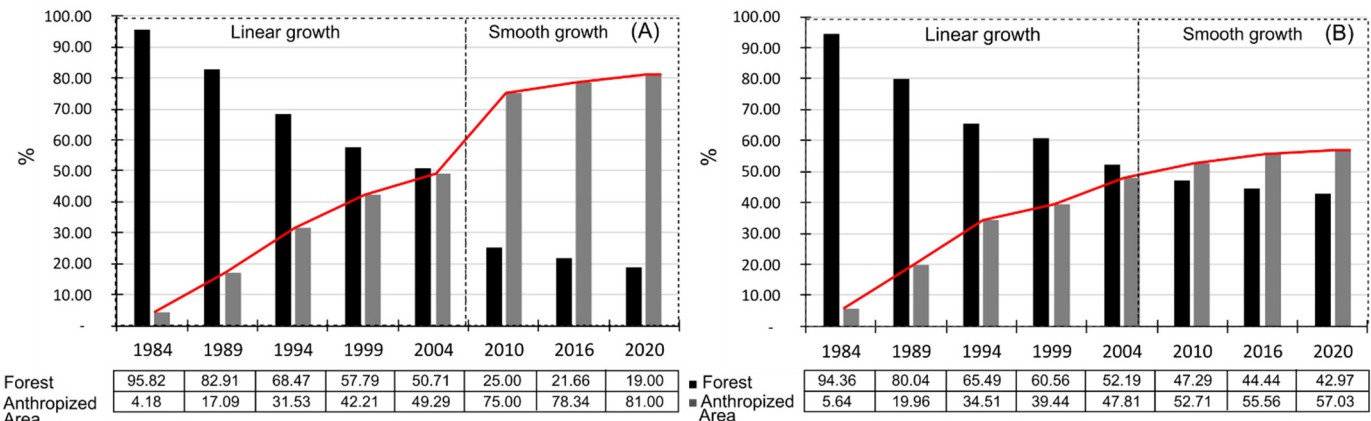

**Figure 4.** Evolution of anthropized areas in the fishbone (**A**) and topographic or disorganised multidirectional models (**B**).

**Figure 5.** Curve representative of the forest fragmentation (**A**) as per average of forest area, (**B**) forest cover and advance of occupation (%) with Secondary Successions (SSA & SSI) (**C**), pasture (**D**), bare soil and agricultural (**E**) areas in the settlements of Machadinho and Anari (1984 to 2020) and Deforestation rate (% of loss forest/year) (**F**).

### 3.2. Paths and Trends of Colonisation

The differences presented by each form of colonisation implemented in the study region, i.e., without (Figure 6A) and with (Figure 6C) concerns regarding the preservation of natural resources and the delimitation of the community reserves, indicates that more naturally friendly colonisation models can be implemented. The methodology discussed focused on three forms of occupation, forest, secondary succession (initial and advanced) and production (pasture, agricultural areas, and bare soil). The results obtained are presented in Table 4, where it is possible to observe the transition between the different uses within the two models analysed in this research.

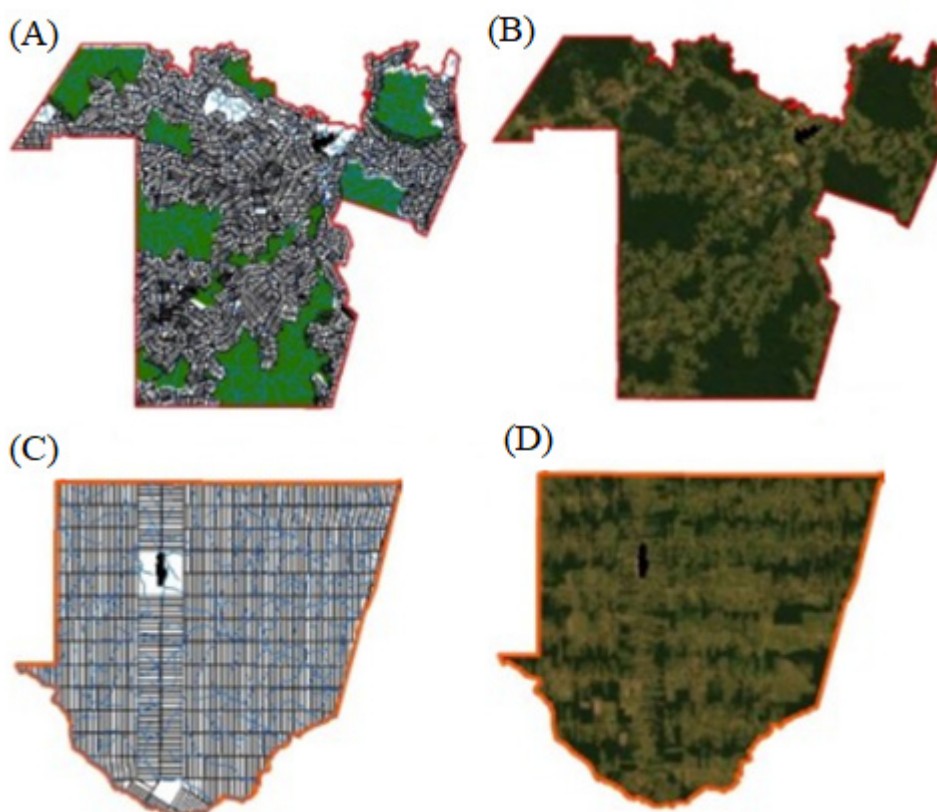

**Figure 6.** Different models of occupation called topographic or disorganised multidirectional (**A**,**B**) and fishbone (**C**,**D**) implemented in Machadinho (**A**,**B**) and Anari (**C**,**D**) settlements.

The transition matrix for the period analysed (1984 to 2020) showed the areas which at the beginning of the study were occupied with forest; 42.97% and 19% remain in the class of origin (forest), respectively for Machadinho and Anari, that is, 57.03% stopped being forest in Machadinho and 81% moved to another class in Anari. Forest to agricultural crops occurred only 3.42% in the topographic model and 8.10% in the fishbone model. This is due to limitations imposed by environmental characteristics that hinder the entry of the first crop. The most common is the exchange from Forest to Pasture, as observed in both models, being higher in the topographic model 43.92% against 39.69% by the model presented fishbone. This common exchange was primarily due to the lack of infrastructure in the topographic model, which generates disorder to explore with other crops. Interestingly, forest class is associated with dynamic changes to the class "SSA" that occurred more frequently in the topographic model, mainly due to non-selective logging (predatory) in reserve areas. Additionally, the biggest difference between the models is observed from Forest to Bare Soil with 9.12% and 32.4% for the topographic and fishbone models, respectively.

**Table 4.** Transition matrix for the fishbone (Anari settlement) and topographic or disorganized multidirectional (Machadinho settlement) models from 1984 to 2020.

|  |  | Forest (%) | Bare Soil (%) | Agricultural (%) | Pasture (%) | SSI (%) | SSA (%) | Total (%) |
|---|---|---|---|---|---|---|---|---|
| Forest | M | 42.97 | 9.12 | 3.42 | 43.92 | - | 0.57 | 100 |
|  | A | 19.00 | 32.40 | 8.10 | 39.69 | - | 0.81 |  |
| Bare soil | M | - | 7.98 | 55.21 | 34.05 | 2.76 | - | 100 |
|  | A | - | 9.96 | 58,53 | 30.61 | 0.90 | - |  |
| Agricultural | M | - | 66.85 | 4.5 | 27.70 | 0.95 | - | 100 |
|  | A | - | 78.22 | 2.23 | 18.58 | 0.97 | - |  |
| Pasture | M | - | 11.99 | 32.44 | 29.47 | 26.1 | - | 100 |
|  | A | - | 12.0 | 9.33 | 55.56 | 23.11 | - |  |
| SSI | M | - | 23.25 | 7.44 | 54.88 | 6.99 | 7.44 | 100 |
|  | A | - | 55.1 | 4.75 | 30.4 | 5.0 | 4.75 |  |
| SSA | M | 7.56 | 2.83 | 4.72 | 73.66 | 5.67 | 5.56 | 100 |
|  | A | 2.78 | 3.7 | 7.4 | 72.2 | 6.48 | 7.44 |  |

M—Machadinho (Topographic or disorganised multidirectional model) and A—Anari (fishbone model).

Pasture class showed that a large percentage of disturbed areas are abandoned, and the initial successional process begins (commonly called "juquira"). This abandonment is more common in the fishbone model than in the Topographic model. In the fishbone model, deforestation is faster (Figure 5F) and is carried out without any concern for the environment. The conversion of pasture to agricultural areas showed a significant increase, especially with rice, maise and soybean crops. This change occurred in 32.44% for the Topographic or disorganised Multidirectional model and 9.33% for the fishbone model.

In the secondary succession, we verified that for each 100ha area that was classified as Initial Succession Stage (SSI) only 23.25% was converted to exposed soil in the model, 54.88% is converted to pasture and 7.44% to agriculture, causing a negative impact of 85.57% on SSI areas, thus preventing progress in the forest regeneration process. It should be noted that only 7.44% of areas in the SSI stage advance in the regeneration process (SSA). This advance occurs mainly in soils of low fertility and also in mountainous areas. The fishbone model results show that 55.51% of the SSI areas are converted to bare soil, 4.75% to agriculture, and only 30.4% to pasture. In the fishbone model, the negative impact on the SSI class was 90.25%. The positive impacts were only 9.75% in the transition from SSA to SSI.

The negative impacts on the secondary succession process in the Amazon Forest are more concerning in the SSA class, where the rates of this impact were 88.77% and 86.08%, respectively, for the topographic and fishbone models. Using a transition matrix (Table 4), it is possible to verify that 73.66% (Machadinho) and 72.2% (Anari) of the areas in the SSA stage were converted to pasture in both models. This transformation is basically due to the fact that these areas were mapped as anthropogenic areas until the year 2008. This particularity, associated with the appreciation of lands and the advance of agriculture and pasture in the Amazon region, justifies the high rates of negative impacts in the study area; Refs. [39–41] (Yanai et al., 2020; Sanchez et al., 2020; Neves et al., 2020) found that this transition scenario is taking place in other areas and regions of the Brazilian Amazon.

Analysing the transition matrix (Table 4), it is noted that the main diagonal presents the values that at the beginning of the occupation process (year 1984) belonged to class a, and at the end of the analysis (year 2020) remained in class a (class source). Still looking at Table 4, it can be seen that the two colonisation projects analysed present a rotation in relation to land occupation, with the original culture. It is also noticed that the pasture class was the class that most changed. That is, it advanced over areas belonging to other classes, highlighting the advance over areas of the SSI and SSA classes.

There is a more dynamic transition process between classes forest, SSI, SSA, pasture, agriculture, and bare soil. The paths between these classes are associated with the cleaning

cycles of vegetation, degradation, or recovery areas. The forest can be changed to bare soil, which can be converted to pasture or agriculture. This can degrade and change to SSA. Regarding the pasture, its dynamics can have different meanings, except for the SSA so that it reaches this stage, you should go through the SSI and then turning into poultry and later to the stage SSA. Areas with pastures follow a different dynamic; they depend directly on agribusiness, in other words, the market will determine if grazing continues or whether it is converted to agriculture and recently bare soil for reforestation. The bare soil can be converted into pasture, converted to agriculture, or abandoned, turning SSI and SSA. The early SSI can be converted into pasture (needing a cleaning) and can be converted to agriculture or bare soil, or even go through the process of succession, hitting the stage SSA. According to the analysed dynamic, the SSA stage is converted to forest, generating a positive impact. This stage can be converted (after deforestation) into production classes (bare soil, pasture, and agriculture). This conversion step is only allowed with authorisation from environmental agencies (Environmental Law). All these changes in the dynamics of land cover and use directly depend on agribusiness. Other factors may influence the trajectory of the dynamics of land use and occupation. The factors were cited by the settlers during the questionnaires and are as follows: biophysical characteristics of the environment, infrastructure (road and urban), credits and incentives, among others relevant to the system. Figure 7 and shows the trajectory illustrative of the landscape dynamics that occur in Machadinho and Anari. An interesting fact related to forest class is associated with dynamic SSA class changes that occurred most frequently in the topographic or disorganised multidirectional model, mainly due to non-selective (predatory) exploration in reserve areas.

The transition dynamics shown in Figure 7 shows that these changes in the landscape produce positive (green line) and negative (red line) impacts [42–44]. The positive impacts were those originating from the vegetation successional process, mainly characterized by the abandonment and/or degradation of pasture [45,46], from which we observe the evolution of the degradation class to SSI, from SSI to SSA and from SSA to forest. The main characteristics verified were the presence of herbs (0.5 m $\leq$ h < 5.0 m) in the degradation class; in the SSI class there was the presence of established seedlings, sticks, and small shrubs (h $\geq$ 3.0 m and $\geq$5 cm; DBH < 10 cm); in the SSA class, there were small, medium, and large trees (10 cm $\leq$ DAP > 15 cm) [47]. The SSI stage began after two years of abandonment of the area, due to the function of inadequate pasture management (e.g., overgrazing) and its duration, ranging from five to seven years. In this study, a time of seven years was adopted to differentiate the change from the SSI stage to the SSA stage, according to [48], who verified as the main characteristic presented in the SSI stage the large presence of individuals with the same species (pioneer vegetation), showing that the topographic model had a greater number of individuals. The main positive impacts observed in the study were the abandonment of the area, the advance in the forest succession process and the maintenance and preservation of forest fragments.

The negative impacts were those that arose through the deforestation of native vegetation with human interference in the successional process [49,50]. This impact was verified between the forest -> deforestation -> pasture -> bare soil -> agriculture transition, and between the SSI -> bare soil or SSI -> agriculture class. Negative impacts from the SSI class occurred due to human intervention in the succession process. This type of impact had been more frequent in areas whose topography provided for agricultural exploitation (commodities). The main negative impacts were scored as deforestation, increased fires, siltation of rivers, changes in the rainfall cycle, visual impact on the Amazon Forest, occupation of areas unsuitable for use.

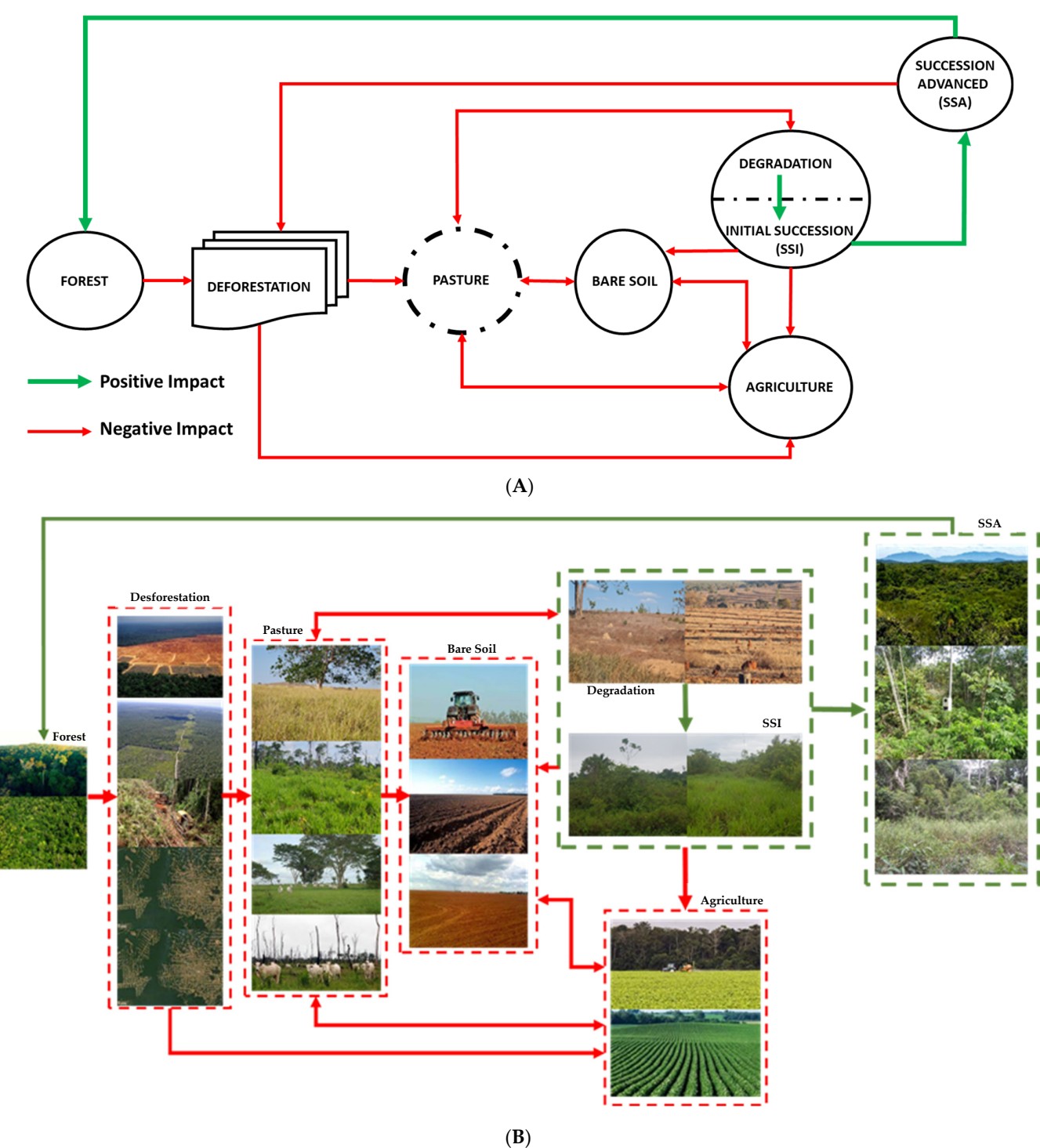

**Figure 7.** Path dynamics of land use and occupation in the Machadinho and Anari settlements corresponding to the occupancy model denominated fishbone and topographic or disorganized multidirectional, average from 1984 to 2020. The red arrows represent the anthropization of native vegetation (forest), causing negative impacts; the green arrows, on the other hand, demonstrate the emergence of secondary vegetation (secondary succession) characterizing the positive impacts for the regeneration of the forest class. (**A**) LULC dynamic and (**B**) LULC dynamics illustrated. Degradation was defined as a function of the characteristics found in loco. Where each class of interest had particular characteristics. For example, in the pasture area, this is considered degraded when it reduces the carrying capacity.

Causes, consequences, and responses in relation to landscape transformation and land use preference in these two settlements are close geographically, but with very peculiar characteristics. Thus, we highlight the design of the models used in the Anari and Machadinho projects, which contribute positively to the increase in negative impacts (change in the landscape), making the road network the main facilitator of these changes. It should be noted that the Anari project features an orthogonal road network and the spatialization of grid-shaped lots, without taking into account the topography and the hydrographic network. In Machadinho, these variables are taken into account to allocate infrastructure resources and common forest reserves.

*3.3. Settlers Responses*

Seeking to understand the people who acquired land within the settlements under-study, a questionnaire was developed that addressed different information (Appendix B).

During the interviews, it was possible to verify that the desire of the settlers in the Anari project did not have a conservationist vision. That is, they wanted to supplant the entire forest, preserving only the permanent preservation areas (riverbanks); unlike the settlers in the Machadinho project, who demonstrated that they had better training and awareness about the preservation of environmental reserves.

The answers obtained were grouped and tabulated; with this, it was possible to define and highlight the agents involved during the colonisation process. For both models, the main difficulties encountered were the biophysical characteristics of the environment, representing 42.5% of the answers, the question of the need for infrastructure, representing 32.7% of the answers, followed by access to rural credits (financing), representing 19.5% of responses, and government incentives, which were highlighted in 5.3% of respondents. In addition, in the Anari project, the low fertility of the soil and questions of how to implement the colonisation model, which often had a wide river, making it challenging to explore the area, were highlighted as difficulties.

## 4. Discussion

The different forms of colonisation, fishbone used in Anari (Figure 6A) and topographic or disorganized multidirectional model used in Machadinho (Figure 6B), presented a diversified LULC dynamic and occupation that can provide support for a series of discussions on the colonisation models used in the BLA, a discussion applicable to any tropical region of the planet. The deforestation process in the study area is associated with the road network, also observed by [13] that examined changes in the large-scale patterns of the main agricultural land use practices in the Brazilian Legal Amazon (BLA) between 1960 and 2013, and [51] observed the same influences of the road network in the process of deforestation in Mato Grosso state, also part of the BLA. Garret et al. [52] stated that access to land is a very important variable in studies of land use and occupation in the Amazon region. For this reason, road openings are often associated with the evolution of anthropized areas and land colonisation. However, the establishment of road systems in the region is a very complex process, varying regionally and locally. On a regional scale, road construction programs have been an important element in the occupation of the Amazon.

It is well known that not only the Anari settlement but the state of Rondonia used the fishbone model throughout the colonisation process. The reasons that led to these choices were explained by [53–55]. In this study, the fishbone model was shown to be more aggressive when compared with the topographic or non-directional model. By enlarging the study area, it is possible to prove the aggressiveness and the power of anthropization caused by this form of occupation.

There is a dynamic transition process between the forest, SSI, SSA, pasture, agriculture, and bare soil classes. The paths between these classes are associated with clearing cycles of vegetation, degradation, or recovery areas (Tables 3 and 4, and Figure 7). The forest can be changed to bare soil, which can be converted to pasture or agriculture. Agriculture

can degrade and change to SSI. In relation to pasture, its dynamic can have different meanings, since this class can become bare soil, agriculture, or SSI, because it depends directly on the agribusiness, that is, the market will determine if the pasture continues or if will be converted in agriculture or to soil, which can be converted into reforestation. The exposed soil can be converted into pasture, which can be converted into agriculture, or can be abandoned, becoming SSI and/or SSA. Early SSI can be converted to pasture (requiring cleaning) and can be converted to agriculture or exposed soil, or even go through the process of succession, reaching the SSA stage. The SSA can become mature forests or can be converted to pasture or agriculture, or even to bare soil. All these changes in the dynamic of land use and coverage depend directly on agribusiness. Other factors may influence these trajectories, being mentioned by the settlers during the interviews, biophysical characteristics of the environment, infrastructure, credits, and governmental incentives among others relevant to the system.

Looking at Figure 7, it is important to note that the dynamics between the "agricultural crops" and "SSI" classes is so sporadic and does not remain for long periods. The dynamics of the "forest" class to "SSA" occurs through the exploration model, very common in areas defined as a legal reserve. The dynamics of the "SSA" class to production classes (agricultural areas, bare soil, and pastures) have occurred recently with great speed due to the large amount of price speculation because areas in the region are due legalisation based on the 2008 moratorium.

Among the classes of production (bare soil, pasture, and agricultural areas) there is a great turnover in the class dynamics, defined mainly by the market. This dynamic occurs in both models, being more evident in the fishbone model. The authors in [56–58] verified that the LULC process in the Amazon region is well consolidated (forest -> deforestation -> pasture -> degradation -> regeneration -> forest). However, with the advance of agriculture, it was possible to verify a change in the trajectory of the landscape dynamics (Figure 7).

As specific details of the dynamics for the different trajectories are observed, as described above, some situations, for example, a steeper or smoother deforestation (Figure 4), not only explains the specific paths to change in the dynamics of land use and land cover, but also sets trends for the near future. In 1989, for example, the agriculture class had only annual crops and young perennial crops (coffee, rubber, and cocoa) [59]. At this early stage, the SSI class arose mainly due to the abandonment of property by the settlers, which did not occur in the SSA class. Despite the main trajectories of land use and land use dynamics, some other trends deserve attention, particularly in agricultural production. In general, trends are represented by the experimentation of different cultures. These crops have significant economic returns but also some uncertainty related to market demands. All the processes associated with the dynamics of occupation and land use in this research directly affected the transformation of the landscape in Machadinho and Anari, that reflexed many other areas in the Amazon and can provide insights for decision making in any tropical area [60]. However, time-series analysis per se is not enough to understand an approach on the causes and effects of this complex process of occupation. The spatial patterns of the landscape are constantly changing, making it difficult to understand. For example, since 1989, the area deforested has increased significantly (Table 3) and [9], due to increased federal government incentives, real estate speculation, and lack of enforcement of environmental laws. The comparative analysis presented here provided a better picture of the heterogeneity of the dynamics of use in the BLA, particularly in areas of rural settlements with different implementation architectures.

It is noteworthy that in the topographical or disorganised multidirectional model, extractive reserves (RESEX) were defined that play a fundamental role in the preservation and conservation of primary vegetation within the settlement. As noted during the study of this time series, both models have suffered and are suffering from changes in the landscape. Currently, greater pressure is occurring in relation to the advance of agricultural crops (soybeans and corn) in the Legal Amazon. Thus, the importance of RESEX has once again been highlighted in the preservation and conservation of forest fragments, unlike the

fishbone model, which has its native vegetation (forest) gradually being decimated and replaced by pastures and agriculture.

In 2020, the population of Machadinho was 40,867, with 59% living in rural areas while Anari had 9384 inhabitants, 81% in rural areas. The average population growth between 2000 and 2020 was 79.72% in Machadinho and 53.80% in Anari. Another heterogeneous aspect between the two areas is that Machadinho has a more structured economy and better commercial and public infrastructure than Anari. The main economic activities of small landowners are sustainable agriculture, commodities, and livestock. Medium and large producers produce beef for local and international markets. The sale or abandonment of land by smallholders is related to lack of subsidies, aging and migration of descendants. As for the economic parameters, the two models presented similar data. According to the last census carried out in 2017 [60], activities related to agriculture accounted for 29.67% and 34.76%, respectively Machadinho and Anari. In relation to per capita income, Machadinho presented R$16,423.33 and Anari R$14,505.19.

The Machadinho and Anari settlements showed different population growth (Figure 8) throughout this series of analysis. One of the main causes was the greater number of settled families and the availability of areas.

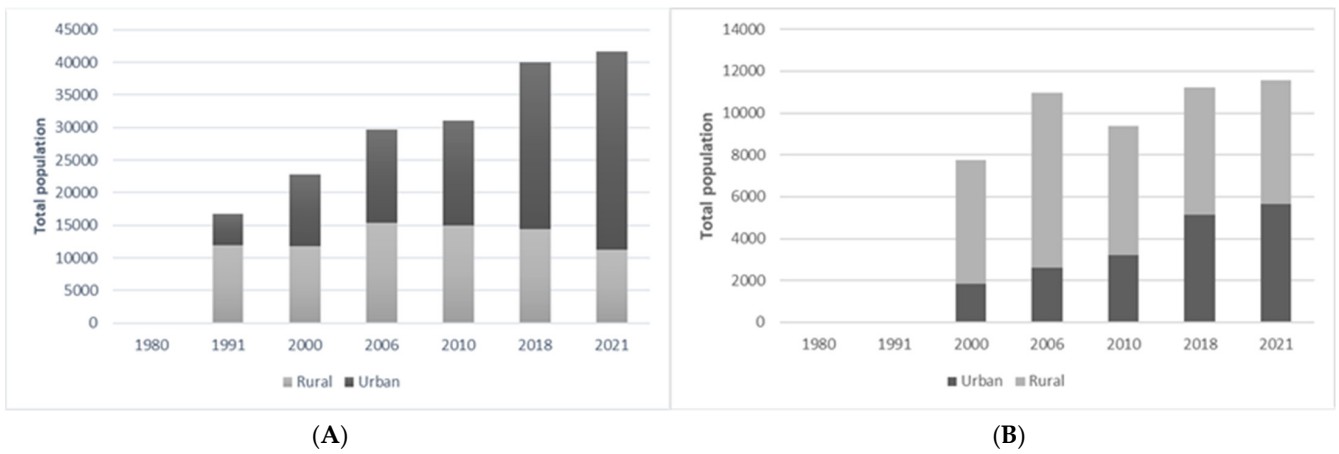

**Figure 8.** Population growth in urban and rural areas, (**A**) Machadinho settlement and (**B**) Anari settlement [61].

Analysing Figure 8, it can be seen that the Machadinho settlement (Figure 8A) presents a more accelerated urbanization trend when compared to the Anari settlement (Figure 8B). In 1991, the Machadinho settlement had approximately 71% of the population living in rural areas. In 2000, a rural exodus (migration to the urban area) began, causing, in 2010, the urban area to represent around 52% of the municipality's population. Currently, the urban population in Machadinho corresponds to approximately 73% of the total inhabitants of the municipality.

The rural exodus that has been taking place in Machadinho is associated with the lack of incentives for small producers, the advance of areas occupied with commodities (soy) and the search for better income conditions and offers of better jobs.

While in the Anari settlement (Figure 8B), there has been an increase in the urban population over the decades, which currently corresponds to approximately 50% of the total population of the municipality.

## 5. Conclusions

The 36 years (1984 to 2020) analysis based on Landsat imagery showed how two different occupation models (without environmental concerns—fishbone, and with environmental concerns—topographic) developed regarding its land use and land cover for each study area. There was a reduction in the areas in the process of secondary succession in Anari due to land price increase after 2010. Nevertheless, in both occupation models, a rapid loss of primary and secondary forests (deforestation) was observed with heavy

dependence on road construction and mainly due to the Agriculture/Pasture relation. This mainly established an understanding of these two forms of occupation and can help future colonisation/occupation programs of the Brazilian Legal Amazon and any tropical rainforest across the globe.

The deforestation rates presented in each model analysed were mainly influenced by the roads, due to the original architecture, as the fishbone model had higher deforestation rates, directly influenced by the distribution of lots inside the settlement.

It was found that in the period from 2016 to 2020, there was accelerated growth in the areas occupied by agricultural crops, mainly soybean crops. It was noted that the areas occupied with agriculture were previously used as pasture and/or in the initial and intermediate process of forest succession.

The analysis of this time series (1984 to 2020) demonstrated the importance of community extractive reserves (RESEX) delimited and materialised in the colonisation model implemented in the Machadinho project, considering that these reserves guarantee the minimum forest cover and function as areas of conservation and protection of biodiversity (the topographic model).

All processes associated with the dynamics of occupation and use of land directly affected the transformation of the landscape in Machadinho and Anari. However, it is noteworthy that only the analysis of the series changes does not allow the analyst to build an approach on the causes and effects of this complex occupation process. This is because the landscape's spatial patterns are constantly changing, making it difficult to understand.

**Author Contributions:** All authors have made significant contributions to this manuscript. V.M. saw the need and relevance of analysing the colonisation models implemented in Rondônia; V.M. and R.d.A.d.S. designed the experiment and performed all field collection and data processing; V.M. and J.A.J. discussed and defined the appropriate methodological procedure; E.M. and V.M. carried out a search in the digital databases of orbital data and selected the best dates to serve as a basis for the study; J.R., R.d.A.d.S. and V.M. wrote this manuscript together. All authors have read and agreed to the published version of the manuscript.

**Funding:** This research was supported in part by the Foundation for Research Support of Rondonia (FAPERO), through Public Notice 011/2018 (AP-INTEC/AGRITECH-FAPERO-Public Notice 011/2018) and by the Federal Institute of Rondonia (Public Notice No. 012/2019/REIT-PROPESP/IFRO and 014/2019/REIT-PROPESP/IFRO).

**Data Availability Statement:** The data presented in this study are available on reseanable request from the corresponding author. The data are not publicly available due to privacy restrictions.

**Acknowledgments:** We would like to thank the Remote Sensing and Geoprocessing Laboratories (GREES) of the Federal Institute of Rondonia for providing the equipment used in the fieldwork and data processing, and the GeoScience nucleus of the State University of West Parana, Cascavel campus (UNIOESTE), for providing the software for data processing and support for on-site work in the research area.

**Conflicts of Interest:** The authors declare no conflict of interest.

## Appendix A

The five transition matrices represent the probability of change among land cover categories recorded for each period analysed in the research.

Legend: A = Agriculture; P = Pasture; SE = Bare Soil; FO = Forest; SSI = Initial Secondary Succession; SSA = Advanced Secondary Succession; IR = Infrastructure and $H_2O$ = Water.

## Appendix B

Land use type: 1-Mature Forest (1.1-Upland; 1.2-Floodplain; 1.3-Open); 2-Savanna (2.1-Woodland; 2.2-Herbaceous/Shrub); 3-Grassland (3.1-Woody; 3.2-Herbaceous/Shrub); 4-Secondary Succession (4.1-Advanced; 4.2-Intermediate; 4.3-Initial); 5-Agricultural Land

(5.1-Perennial; 5.2-Agroforestry; 5.3-Annual; 5.4-Pasture); 6-Barren Land (6.1-Agricultural Exposed Soil; 6.2-Non-Agricultural Exposed Soil); 7-Builtup Land (7.1-Road; 7.2-Urban Area); 8-Water. @ Technology: Manual-MAN or Mechanized-MEC. * Inputs: Fertilizer-FER; Lime (calcareo)-LIME; Ash-ASH.

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
