# Peer review of "Three Decades after: Landscape Dynamics in Different Colonisation Models Implemented in the Brazilian Legal Amazon"

_remotesensing, doi:10.3390/rs13224581_

Round 1

Reviewer 1 Report

Review on remotesensing-1429457

This paper compared the two kind of land use/cover changes (LUCC) happened in the Brazilian Legal Amazon through Landsat-based remote sensing data applying Random Forest algorithm for the period since 1984 to 2020 and the changing process and reasons were attributed to expanding of agriculture, pasture and roads. The authors tried to provide a methodology that effectively contributes to evaluate multitemporal dynamics of land use and coverage through regional-scale analysis for the different projects. As one of topics in global changes, LUCC is a very interesting problem, especially over the tropical forest in a faster decreasing of its area.

However, I have some major concern on this study.

Firstly, it would mislead on the trajectory of LUCC over the two sites. As a suggestion, the authors need to consider the background of the two sites, especially their roads design and population growth. From the figure 1, those two sites showed very contrast roads map that would be a result of human design. Population growth, natural environmental condition, friendly social-economic environment, as discussed around lines 541 to 552 in the manuscript, would be more important to format the two trajectories. But the inter-annual changes of population growth were not showed, which results in unclear relationship between them.

Secondly, on the view of methodology, this study did not show any background on the methods. In the introduction, we cannot find any words on the methods review on the remote sensing-based land use/cover classification, or random forest algorithm, or methods evaluating multitemporal dynamics of LUCC through regional-scale analysis for the different projects. Because the most readers would be more interested in the methods, except readers concerning tropical forest changes.

Thirdly, as a LUCC-based, not a method-based manuscript, it would be not very good idea publishing on this journal, Remote Sensing.

Some minor problems:

  • The abstract uses more words on the background but is not clearly to explain the scientific question. Why those two models are important to being evaluated? And what this manuscript could be referred by readers.
  • The introduction, the first paragraph, as my opinion, is not relative with the topic, except the last sentence.
  • Study area, the second paragraph, when the roads were designed and constructed? More additionally, population should be explained in this part.
  • Section 2.2, suggest to introduce data source and their process. I concerned the data quality and data process, therefore, please show the satellite image data after process and used as inputting for random forest algorithm.
  • Section 2.2, the two classes, SSI and SSA, please explain their definition and measurement. The class, bared soil, looks like farm land in non-growth season according to the figure 7b, is it? So the bared soil should be defined by some conditions, yet.
  • In figure 1, there are too much fishnet lines without its legend in the Anari map, is there any necessary? Same question in figure 6C.
  • In figure 7b, how did you define “degradation”? what’s difference with the seasonal landscape, such as harvest or dry season?
  • Lines 477, please give a summary on the reason.
  • Conclusion, as my opinion, why did not consider population growth as an essential factor? Those two areas have very different population in 2020 according the description in the lines 541 to 543. But unfortunately, the changes of population were not shown in this manuscript.

Reviewer 2 Report

Excellent research. I would suggest to use same north arrow and scalebar style for all of your maps. The manuscript lack in consistency which can be improved. I would suggest to use tables to minimize written description.

Round 2

Reviewer 1 Report

The authors well responded those comments that would be concerned by readers but were not found in the revised manuscript, for example, the concerns on the data quality, the definition on SSI and SSA, the degradation. It was suggested to revised in MS not only responded them to reviewer.

Therefore, this version still needs some minor revisions.

Author Response

Requests made by reviewers were met, in the manuscript. In order to facilitate the visualization, it is written in the manuscript in red color.

Thanks to the reviewers for the suggestions for reviewing the text. Your suggestions made the article more robust and better to understand. Thanks.

“Comments and Suggestions for Authors

The authors well responded those comments that would be concerned by readers but were not found in the revised manuscript, for example, the concerns on the data quality, the definition on SSI and SSA, the degradation. It was suggested to revised in MS not only responded them to reviewer”.

R=

  • Characteristics of the models, described in lines 90 to 95;
  • Contextualization of the SSI and SSA classes, described in lines 141 to 150 and,
  • Contextualization of degradation, described in lines 432 to 434.

This manuscript is a resubmission of an earlier submission. The following is a list of the peer review reports and author responses from that submission.

Round 1

Reviewer 1 Report

Goal of the paper is not clear. Describe it.

I do not see any basic questions, which can be tested.

Describe differences of environmental conditions between Machadingo and Anari.

Methods are not clear. The flowchart (fig. 2) must be related to yours goals.

Transition matrix analysis is suitable method (lines 128-143), but results are completely missing!

Eq. 1 is probably wrong because division by final forest area can lead to division by zero in some cases.

Methods on the questionnaire goals, structure and processing is missing.

Deforestation is related with very important fragmentation of natural forests. Calculate changes in the measure of fragment size (e.g. average of area, length of forest edges per unit of area, ...).

The linear growth behaviour (line 198) is only approximation, not in sense of statistics.

The sentence starting at line 206 is not true: highly distinct values in fig. 5a at 1984.

Table 4 does not represent transition matrix because information on size of stable category (e.g. forest-to-forest) is missing.

Deforestation represents current process in whole Amazonia. Compare your results with common situation (in Brazil, ...). Pay attention to the problems of deforestation in nature conservation point of view.

Author Response

Dear editor,

Thank you very much for the opportunity to revise the document, the comments and suggestions provided by the reviewers greatly improved the manuscript. Thus, we are grateful for their inputs. We revised the manuscript, addressing all the suggestions and comments of both reviewers. A native speaker revised the English, and all edits are in red. In order to facilitate the edits check, please find below the responses in blue to the reviewers’ comments indicating how we modified the document.

Kind regards,

Valdir Moura

Reviewer #1

Goal of the paper is not clear. Describe it. I do not see any basic questions, which can be tested.

R: It is hypothesised that a nature-friendly colonisation model can be more sustainable over time when compared to a colonisation model that disregards the surrounding environment. Thus, a comparison between a topographic (in Machadinho) and a fishbone (in Anari) colonisation models was performed.  We added such a sentence in the last paragraph of the document clarifying the paper’s goal.

Describe differences of environmental conditions between Machadingo and Anari.

R: The locations are neighbouring municipalities and share the same environmental conditions. A sentence was added in section 2.1 to clarify this comment. This makes them perfect for such study as essentially two areas with the exact same ‘start’ under different colonisation schemes.

Methods are not clear. The flowchart (fig. 2) must be related to yours goals. Eq. 1 is probably wrong because division by final forest area can lead to division by zero in some cases. Methods on the questionnaire goals, structure and processing is missing.

R: The whole methods section was rewritten more straightforwardly, also incorporating Reviewer #2 comments. There was some translation confusion as well that did not help the clarity of the method. Equation 1 typo was fixed; the division is by the initial area, not the final, following FAO guideline for forest survey [manuscript reference 35 and 36]. Questionnaires were added in appendix B.

Transition matrix analysis is suitable method (lines 128-143), but results are completely missing!

R: The transition matrix from 1984 to 2020 was added as Table 4.

Deforestation is related with very important fragmentation of natural forests. Calculate changes in the measure of fragment size(e.g. average of area, length of forest edges per unit of area, ...).

R: As a measure of fragmentation, we calculated the average forest area as requested and added the plot in Figure 5.

The linear growth behaviour (line 198) is only approximation, not in sense of statistics.

R: We changed to proxy-linear growth behaviour, thus not in the sense of statistics.

The sentence starting at line 206 is not true: highly distinct values in fig. 5a at 1984.

R: We found an error in Figure 5. There was no agriculture or bare soil before settlement. We also updated it to add the fragmentation indicator as requested by the reviewer.

Table 4 does not represent transition matrix because informationon size of stable category (e.g. forest-to-forest) is missing.

R: Table 4 was fixed.

Deforestation represents current process in whole Amazonia. Compare your results with common situation (in Brazil, ...). Pay attention to the problems of deforestation in nature conservation point of view.

R: A paragraph was added to the discussion. The common situation in Rondonia is the fishbone settlement strategy. This paper highlights the needed change, showing that the topographic settlement can reduce the human impact in the forest in the long term.

Reviewer 2 Report

Review: “Three decades after: landscape dynamics in different colonization models implement in the Brazilian legal amazon

This study presented a multi-temporal land use land cover change analysis in two areas in Brazilian Amazon. The topic is interesting and fits well in the scope of the journal. However, The methods sections 2.2, 2.3, and 2.4 are disorganized and missing important information.

In section 2.2 Methods, it mentioned image segmentation without telling for which purpose it was used. It also did not introduce images that were segmented. It then gave two sets of very different parameters for images of TM and OLI without explaining how the parameters were derived and why they were so different. In a similar way, it mentioned the classification was done by random forests classifier without telling how the classifier was parameterized. There was no explanation for the connection between image segmentation and random forests classifier.

In section 2.3, about Data and LULC mapping, here it repeated the classes that were already mentioned in section 2.2 but used different names for a couple of classes which caused confusion. It briefly mentioned the number of training samples (380 samples each class) without telling how the training samples were derived, if they split the samples into training and validation, and how about the number of trees and the predictor variables. For the accuracy assessment of the LULC classifications, no information provided about the sampling scheme and how the assessment was carried out, for example if they used confusion matrix.

Section 2.4, LULC dynamics analysis, in the section, there are sentences that do not make any sense, for example “Each probability matrix was then separated so that the vectors and eigenvalues could be calculated as presented by [33]”. I am not sure what those vectors and eigenvalues were, please explain. Another one is “These matrices were used to simulate the proportion of cover that could stabilize the deforestation.”  Here please explain the meaning of “stabilize the deforestation”. I recommend the authors could carefully check the manuscript and make sure these are mended.

In the results, line 161-163, “During the initial phase of implementation of both settlement projects (base year 1984), there was a similar percentage of forest (95%) and deforestation (5%).” It is not right to equate the non-forest area as deforestation. Deforestation is a land cover change, which implies a comparison of the forest in time 1 with time 2 though a transition matrix, and deforestation is the change from forest to non-forest area. By looking at land cover map of one time, you can not know deforestation. Figure 4 is therefore erroneous since it is mistaken non-forest area in each date as deforestation. Also, no transition matrix was presented in the results section, I wonder how the LULC change was computed.

I recommend the authors to reorganize and rewrite the methods and results section, and carry out more analysis for this research.

Author Response

Dear editor,

Thank you very much for the opportunity to revise the document, the comments and suggestions provided by the reviewers greatly improved the manuscript. Thus, we are grateful for their inputs. We revised the manuscript, addressing all the suggestions and comments of both reviewers. A native speaker revised the English, and all edits are in red. In order to facilitate the edits check, please find below the responses in blue to the reviewers’ comments indicating how we modified the document.

Kind regards,

Valdir Moura

Reviewer #2

This study presented a multi-temporal land use land cover change analysis in two areas in Brazilian Amazon. The topic is interesting and fits well in the scope of the journal. However, The methods sections 2.2, 2.3, and 2.4 are disorganised and missing important information.

R: The whole methods section was revised. There were some translation issues, particularly regarding segmentation/classification/mapping nomenclature.

In section 2.2 Methods, it mentioned image segmentation without telling for which purpose it was used. It also did not introduce images that were segmented. It then gave two sets of very different parameters for images of TM and OLI without explaining how the parameters were derived and why they were so different. In a similar way, it mentioned the classification was done by random forests classifier without telling how the classifier was parameterised. There was no explanation for the connection between image segmentation and random forests classifier.

R: the word segmentation and mapping were indiscriminately used due to translation confusion. We clarify this in the methods. The Random Forest was used for mapping. The methodology was revised and the section is more straightforward than the previous version.

In section 2.3, about Data and LULC mapping, here it repeated the classes that were already mentioned in section 2.2 but used different names for a couple of classes which caused confusion. It briefly mentioned the number of training samples (380 samples each class) without telling how the training samples were derived, if they split the samples into training and validation, and how about the number of trees and the predictor variables. For the accuracy assessment of the LULC classifications, no information provided about the sampling scheme and how the assessment was carried out, for example if they used confusion matrix.

R: A total of 2,480 samples were used to train and validate the model in a random 70-30 slipt. That is, 70% (1,736 samples) were randomly selected and used for training the model, and the remaining 30% (744 samples) were used to validate the LULC maps. The common practice confusion matrix and global accuracy, kappa index, and producer accuracy were calculated with the validation set. This section was added to the M&M section to clarify this issue.

The random forest hyperparametrisation was also better explained. Due to the differences in sensor radiometric specifications, different random forest models were used. One for Landsat TM with the number of variables randomly sampled as candidates at each split (mtry) of 12 and the number of trees (ntrees) of 200. And another one for Landsat OLI with mtry of 28 and ntrees = 500. To obtain these parameters, a range of mtry (10 to 30) and ntrees (100 to 1000) were tested. This was added to section 2.2

Section 2.4, LULC dynamics analysis, in the section, there are sentences that do not make any sense, for example “Each probability matrix was then separated so that the vectors and eigenvalues could be calculated as presented by [33]”. I am not sure what those vectors and eigenvalues were, please explain. Another one is “These matrices were used to simulate the proportion of cover that could stabilise the deforestation.” Here please explain the meaning of “stabilise the deforestation”. I recommend the authors could carefully check the manuscript and make sure these are mended.

R: The whole M&M section was rewritten, and these issues were all amended. There was some loss of meaning when translating the document; hence, some sentences were without any sense. A native English speaker carefully checked this issue.

In the results, line 161-163, “During the initial phase of implementation of both settlement projects (base year 1984), there was a similar percentage of forest (95%) and deforestation (5%).” It is not right to equate the non-forest area as deforestation. Deforestation is a land cover change, which implies a comparison of the forest in time 1 with time 2 though a transition matrix, and deforestation is the change from forest to non-forest area. By looking at land cover map of one time, you can not know deforestation. Figure 4 is therefore erroneous since it is mistaken non-forest area in each date as deforestation. Also, no transition matrix was presented in the results section, I wonder how the LULC change was computed.

R: The transition matrix was added as Table 4, and the terminology was improved. Figure 4 was also fixed. The incorrect terminology might be due to translation as in Portuguese, it is common to refer to anthropised areas as ‘deforestation‘.

I recommend the authors to reorganise and rewrite the methods and results section, and carry out more analysis for this research.

R: We followed the reviewer’s recommendation and rewrote both M&M and Results sections. More analysis was carried out following the reviewer’s #1 suggestions.

Round 2

Reviewer 1 Report

The text is better than the first version.

The transition probabilities are in the appendix, but I do not find any statistical evaluation (e.g. eigenvalues, eigenvectors, their interpretation).

Author Response

Response to Reviewer 1 Comments

Dear editor,

Thank you very much for the opportunity to revise the document once again, the comments and suggestions provided by the reviewers improved the manuscript. Thus, we are grateful for their inputs. We revised the manuscript, addressing all the suggestions and comments of both reviewers. A native speaker carefully revised the English, and all edits are in red. In order to facilitate the edits check, please find below the responses in blue to the reviewers’ comments indicating how we modified the document.

Kind regards,

Valdir Moura

Comments and Suggestions

“The text is better than the first version”.

Thanks for the comment. I have been striving to provide the reader with a text that is clearer and with better quality. Your encouragement helps me in seeking this improvement.

We are thankful for the reviewers' suggestions that made the text better.

“The transition probabilities are in the appendix, but I do not find any statistical evaluation (e.g. eigenvalues, eigenvectors, their interpretation)”.

The evaluation (based on eigenvalues and vectors) is summarized in Table 4. Paragraphs were added on page 13 with further interpretation of the transition matrix.

Reviewer 2 Report

The methods and results sections are still very problematic.

Section 2.2

Line 106-108, please indicate the dates on which the Landsat images were obtained for each year.

Line 126, this should result in 8 land cover maps.

Line 128-133, please explain which and how many variables were used for Landsat TM and OLI images classification.

Section 2.3

Line 151, please explain for which time periods you calculated the rate of deforestation and please present the results of deforestation analysis.

Author Response

Response to Reviewer 2 Comments

Dear editor,

Thank you very much for the opportunity to revise the document once again, the comments and suggestions provided by the reviewers improved the manuscript. Thus, we are grateful for their inputs. We revised the manuscript, addressing all the suggestions and comments of both reviewers. A native speaker carefully revised the English, and all edits are in red. In order to facilitate the edits check, please find below the responses in blue to the reviewers’ comments indicating how we modified the document.

Kind regards,

Valdir Moura

Section 2.2

Line 106-108, please indicate the dates on which the Landsat images were obtained for each year.

Knowledge about image acquisition dates is extremely important, and serves to demonstrate the best periods to obtain good quality passive remote sensing products (no cloud). The period used comprises the months of June, July and August, considering that in this interval there are no problems with clouds, which would make studies difficult and also due to the type of use and occupation used in the study region.

In the article, this request was inserted in section 2.2 lines 109 to 112.

A sentence was added in section 2.2 with the image acquisition dates. Now reads:

‘Images were acquired in June, July or August of each year as follows: 1984 (1984-08-04), 1989 (1989-07-17), 1994 (1989-07-17), 1999 (1999-07-29), 2004 (2004-07-26), 2010 (2010-07-27), 2016 (2016-07-11) and 2020 (2020-07-22). The use of the images acquired during this period ensured cloudless images.’

Line 126, this should result in 8 land cover maps.

Correct, that was a typo and now is fixed and matches the resulting maps on Figure 3a and 3b.

The analyzed period originated 8 (eight) annual LULC maps (1984, 1989, 1994, 1999, 2004, 2010, 2016 and 2020), as shown in Figures 3a and 3b. The periods of analysis were (1984-1989, 1989-1994, 1994-1999, 1999-2004, 2004-2010, 2010-2016 and 2016-2020), totaling 7 intervals. These periods gave rise to the transition matrices presented in Appendix A.

The correction has been carried out and is on line 136.

Line 128-133, please explain which and how many variables were used for Landsat TM and OLI images classification.

Bands 3, 4, and 5 were used with TM and bands 4, 5, and 6. This information is stated in lines 107-108.

Section 2.3

Line 151, please explain for which time periods you calculated the rate of deforestation and please present the results of deforestation analysis.

We revised the annual rate of deforestation and found a standardized method proposed by Puyravaud (2003). The results were determined for each interval and are presented in Figure 5. The periods were for each studied interval. This was added in the deforestation rate section.

Puyravaud, J P (2003) Standardizing the calculation of the annual rate of deforestation, Forest Ecology and Management, Volume 177, Issues 1–3, Pages 593-596, https://doi.org/10.1016/S0378-1127(02)00335-3.
